:ֹ᛬֟PLOS ONE

# Monitoring the 'diabetes epidemic': A framing analysis of United Kingdom print news 1993-2013

**Kristen Foley**๏⍟*, **Darlene McNaughton**⍟, **Paul Ward**⍟

Discipline of Public Health, Flinders University, Adelaide, South Australia, Australia

⍟ These authors contributed equally to this work.
* Kristen.Foley@flinders.edu.au

## Abstract

### Introduction

The view that we are in the midst of a global diabetes epidemic has gained considerable ground in recent years and is often linked to the prior 'obesity epidemic'. This research explored how the diabetes epidemic was represented in United Kingdom (UK) news over the same time period that the obesity epidemic was widely reported. The research was motivated by a sociological interest in how postmodern 'epidemics' synergise with each other amidst broader political, economic, moral and sociocultural discourses.

### Method

We analysed three time-bound samples of UK news articles about diabetes: 1993 (n = 19), 2001 (n = 119) and 2013 (n = 324). Until now, UK media has had the least attention regarding portrayal of diabetes. We adopted an empathically neutral approach and used a dual method approach of inductive thematic analysis and deductive framing analysis. The two methods were triangulated to produce the findings.

### Results

Framing of diabetes moved from medical in 1993 to behavioural in 2001, then societal in 2013. By 2001 obesity was conceptualised as causal to diabetes, rather than a risk factor. Between 2001 and 2013 portrayals of the modifiable risk factors for diabetes (i.e. diet, exercise and weight) became increasingly technical. Other risk factors like age, family history and genetics faded during 2001 and 2013, while race, ethnicity and culture were positioned as states of 'high risk' for diabetes. The notion of an 'epidemic' of diabetes 'powered up' these concerns from an individual problem to a societal threat in the context of obesity as a well-known health risk.

### Discussion and conclusion

Portraying diabetes and the diabetes epidemic as anticipated consequences of obesity enlivens the heightened awareness to future risks in everyday life brought about during the

**Data Availability Statement:** The data underlying our study are third party data, however we do not have permission to legally distribute these data within the parameters of copyright law. The data can be accessed by using the search terms and

time periods specified in the methods section of the paper (keywords appear in Fig 1). No special access privileges beyond subscription to Factiva and Lexis Nexis databases are required.

**Funding:** The author(s) received no specific funding for this work.

**Competing interests:** The authors have declared that no competing interests exist.

obesity epidemic. The freeform adoption of the 'epidemic' term in contemporary health discourse appears to foster individual and societal dependence on biomedicine, giving it political, economic and divisive utility.

## Introduction

The view that we are in the midst of a global diabetes epidemic has gained considerable ground in recent years, often asserted to be driven by the prior 'obesity epidemic' [1]. To date there has been limited empirical research that explores how the diabetes epidemic has been constructed in the media and also no longitudinal analysis of how the diabetes epidemic has been positioned in reference to the obesity epidemic. This research aimed to explicate the trajectory of the diabetes epidemic in United Kingdom (UK) newspapers over the time period that the obesity epidemic was widely reported. In the background we explore some key points from the critical literature about presentations of health issues and postmodern epidemics. We then chart the current literature regarding how diabetes has been presented in the news. The method section outlines an innovative paired thematic and framing analysis we developed to answer our research question. In the results section we provide a chronological review of how diabetes and its epidemic was reported in UK news, in relation to reporting about the obesity epidemic. The discussion section explores links between our results with other contemporary critical public health literature.

Understanding popular conceptions of health issues is an important field of public health inquiry, as notions of health and healthiness are imbued with great political, economic, moral and sociocultural value. Critical scholars contend we live in a time fixated on health as a life-long goal, where 'healthy lifestyles' act as a vehicle for people to pursue this goal and simultaneously demonstrate their commitment to this collective priority [2, 3]. The widespread production and consumption of goods and services to achieve healthy living attests to the economic value tied to healthy lifestyles [4]. This extreme focus on health, mobilised through healthy lifestyles, is considered a political tool that neoliberal governance systems use to prevent public health burdens [5].

For some, a societal focus on health and an attention to healthy living can be a positive outcome of public health campaigns and civic governance with wide reaching benefits. However, as the concepts of health and healthiness have become politically and economically important, they have also been imbued with meanings around moral and social worth [3]. An individual's perceived healthiness is often used as a proxy for evaluating the success of social and personal life [2]. Conversely, perceived neglect of one's health can signify a failure to adhere to shared norms and values [6]. Individuals who develop diseases constructed as preventable, such as diabetes or obesity, are therefore portrayed as 'bad' or 'irresponsible' citizens [7, 8]. Others may even be defined as a 'threat' to societal values and interests [7]. Living a 'healthy life' therefore becomes doubly important to avoid shame, blame, exclusion, and myriad forms of 'othering' based on perceived disease probability [3].

Studying popular constructions of health provides valuable insight into discourses around health and healthiness and how they synergise with moral, economic, political and sociocultural meaning. The freeform adoption of the 'epidemic' term to popular presentations about health issues is of great academic interest because it can evocatively tie concerns of individual healthiness to concerns of public safety or progress [7]. The obesity epidemic, for example, is argued to have legitimised concerns about body weight through the powerful narrative of

western medicine. Raisborough [9] asserts that our understanding of health has become 'fat tainted' through the obesity epidemic because it enhanced legitimacy around moral or social concerns around being overweight or obese, by framing fatness as a health issue. Maintaining the use of words like 'fat' or 'fatness', to many authors in critical health disciplines, resists the way in which a bodily state has become so heavily medicalised [10, 11]. Some authors point out that the measurements and classifications used to construct the obesity epidemic was over-stated, while other shifts in postindustrial societies were neglected in its portrayal [10, 12, 13]. Campos [12] highlights that escalating health concerns about obesity ushered in enhanced attention to health risks because everyone, everywhere, is at risk of becoming too fat. This is an example of Lupton's [14] concerns that health risks provide effective political mechanisms to govern a health-conscious public. For this reason, Boero [15] argues that supposed health epidemics must be brought into the realm of cultural critique.

It is from this critical cultural perspective that we must 'monitor' the trajectory of the 'diabetes epidemic' in popular discourse vis a vis the prior 'obesity epidemic'. Until now there has been no empirical nor longitudinal analysis of the diabetes epidemic, despite it gaining considerable ground over the past two decades and converging with fatness in public, scientific and policy arenas [1]. We use the term diabetes in this article to refer predominantly to Type 2 Diabetes Mellitus which is positioned as the driver of the epidemic, and also to reflect the way that diabetes is being constructed in popular discourse. Increases in rates of diabetes are likely to be significantly influenced by an enhanced focus on surveillance and diagnosis in our modern era [1, 16]. Further, ageing populations in industrialised countries likely play a significant role in increasing appearance of the disease, given the strong association between ageing and diabetes [1]. Socioeconomic disadvantage is strongly associated with risk for diabetes [17–23] as well as more serious complications from the disease [24, 25]. Despite a wealth of molecular and genetic data collection over the past few decades still no answer has been found for exactly what causes the epigenetic disease [24, 26–28]. These complexities and uncertainties, however, are not well represented in discussions about the diabetes epidemic [29] while obesity is often emphasized at the exclusion of factors beyond individual control [1].

A handful of studies have explored how diabetes has been portrayed by news media. Rock [30] argued diabetes was portrayed as a sinister medical condition in United States press. Gollust and Lantz [29] showed that obesity and behaviour were dominant features of how diabetes is portrayed in North American news. Hellyer & Haddock-Fraser [31] found UK news poorly reported the social determinants of health associated with diabetes. McNaughton [1] argued that obesity and diabetes have become conflated across realms of media, policy and scientific circles in Australia—and that the so called obesity epidemic is commonly depicted as the primary cause of a diabetes epidemic. In Canada, government, academic and public health campaigns cite behaviour as a central driver of the disease [32, 33] and similarly the obesity epidemic is identified as the key driver of the diabetes epidemic. Most recently in the United States (US), Stefanik-Sidener [34] argued that a behavioural frame was commonly used to describe the disease in the *New York Times*. No studies have yet taken a longitudinal approach to explore the trajectory of diabetes presentations over time. Further, no one has yet empirically analysed the emergence of the diabetes epidemic in news accounts vis a vis obesity and its own epidemic. Critical public health has, however, received the mandate to 'dissect potentially toxic social, economic, political and cultural processes' [35]. It is with this sociological intention we aimed to interrogate the emergence of a diabetes epidemic in popular discourse and its linkages with the obesity epidemic.

In considering the current attempts to explicate how diabetes has been portrayed in the news, the United Kingdom has received the least attention. Hellyer and Haddock-Fraser's [31] work applied content analysis to three months of media coverage, finding that diabetes was

portrayed as a behavioural issue and that obesity was causal to the disease. Obesity was a key policy issue during the late 1990's in the UK and they remain in the midst of an 'obesity epidemic' [36]. Monaghan et al. [37] have drawn attention to England's reputation as the 'fat man' of Europe, and the 1998 WHO Obesity Report [38] specifically named England as having doubled their 'obesity levels' in a relatively short period of time. Diabetes UK identified in 2015 that it is the 'Age of Diabetes' [39]. UK media were therefore considered a useful landscape to understand and contextualise how presentations of diabetes change over time in relation to discursive shifts about fatness. Our research question was therefore:

How do presentations of diabetes change over time in UK news in relation to presentations of obesity and the obesity epidemic?

A critical location in which these discursive tangles operate, and are visible, is within today's media. The reporting of health risks in the media has become a core news business in the past two decades [40], likely reflecting enhanced cultural interest in health and health risks. In a spiraling fashion, news about health is designed to elicit public interest, but in doing so also directs it [40]. Through a cascading network of issue interpretation the media are thought to play a key role in setting public understandings about a range of issues [41]. Others argue they co-construct the structures through which the public understand health and illness [42] and bolster the visibility and validity of contemporary health 'epidemics' [15, 43]. Journalists admit that much of what drives reporting about health is newsworthiness and sensationalism, and that risk is a key way of garnering public attention [44]. While what the media 'is' is constantly changing, their role as an interface with public knowledge remains relatively stable [45]. Media products therefore showcase processes of public discussion, and are a useful platform for exploration of how the diabetes epidemic has evolved over time and in relation to the moral and cultural discursive shifts brought to bear during the earlier 'obesity epidemic'.

## Materials and methods

### Sampling

A selection of newspapers was developed based on their completeness in electronic storage databases Factiva and Lexis Nexis from the early 1990s to 2013 (searching was undertaken February 2014). The three brands that offered high readership (determined via National Readership Statistics) as well as diversity in tone, political orientations and target audience were selected. We included *The Times*, *The Guardian* and *The Daily Mail* (see Table 1 below). The newspaper articles used in this sample are available online via Lexis Nexis and Factiva. We used

**Table 1. Newspaper demographics.**

| Date Range | 1990–2013 | | Newspaper Demographics | | | |
|---|---|---|---|---|---|---|
| Databases | Lexis Nexis + Factiva | | Political Orientation [46] | Tone [47] | Target | Owned By |
| | | | | | | |
| Newspaper | The Times Times on Sunday | | Right (Backed Tories last election; listed in top 3 for anti-labour articles) | Quality | High SES | Subsidary of News UK, owned by News Corp, owned by Rupert Murdoch [48] |
| | The Guardian | | Left (Backed Labour in last election) | Quality *Focus on independent journalism* | High SES | Scott Trust [49] |
| | The Daily Mail Daily Mail on Sunday | | Centre-Right (Conservative/ Liberal Democratic support at last election) | Mid-market tabloid *Focus on women* | Low or Middle SES | Lord Rothmere via 'Daily Mail and General Trust' [50] |

**Fig 1. Keyword searches.**

both Lexis Nexis and Factiva to enhance the likelihood of gaining a complete set of articles published in these three papers and their Sunday editions across our time period of interest.

Keywords were drawn from the extant literature exploring how diabetes and obesity are discussed in the media and in academic and grey literature [1, 29, 31–34, 51, 52]. These keywords were applied to the full text of articles in order to yield as many newspaper articles for analysis as possible (as shown below in Fig 1).

Searching yielded a total of 4280 articles in the 23 years between 1990 and 2013. We decided to isolate three years for analysis to more clearly explicate changes across time in how diabetes was presented in relation to our other search terms. Consequently articles published in 1993, 2001 and 2013 were extracted. Our logic for choosing these years was that the 2001 sample would capture early discursive changes after the obesity epidemic was announced and reported widely in 1997/1998 –three to four years after the increased media coverage. Analysis of the 2013 sample, 12 years later, would showcase whether any of those changes had been retained or naturalised. Analysing articles from 1993 would give a 'baseline' from which to evaluate the changes seen in later samples because it was thought to be pre-obesity era, and was also one of the earliest times at which articles were electronically indexed.

Articles from 1993, 2001 and 2013 were selected electronically in both Factiva and Lexis Nexis and imported into Zotero, where automatic de-duplication was undertaken by Author 2. From Zotero, the articles were then imported into NVivo (v10). The numbers of articles imported into NVivo for each subsample are shown below in Table 2, alongside further duplicates removed by title scan that had been missed by Zotero (n = 7). This secondary, hand duplication was completed solely by Author 1. Full-text articles were then read by Author 1. Articles were excluded only if they mentioned diabetes in passing but did not communicate anything meaningful about how diabetes was conceptualised in relation to the other search terms: for example, one article discussed the practices of a drug company that produced diabetes medications, but no meaningful content around diabetes in relation to obesity was included in the article. Authors 2 and 3 checked all articles proposed for exclusion in 1993; 50% from 2001; and 20% from 2013. The spread of resultant included articles across the three newspaper brands is also detailed in Table 2.

## Positioning

We adopted an approach of empathic neutrality in this research. Patton [53] describes this as a criteria of good qualitative research whereby the researcher must occupy a 'middle ground'

**Table 2. Time-bound samples of newspaper articles.**

|  | Newspaper Articles | | | | Newspaper distribution | | |
|---|---|---|---|---|---|---|---|
|  | Retrieved | Duplicates | Exclusions | Included | *The Times (Sunday)* | *The Guardian* | *The Daily Mail (Sunday)* |
| **1993** | 26 | 3 | 4 | **19** | 6 (0) | 4 | 9 (0) |
| **2001** | 130 | 3 | 8 | **119** | 35 (0) | 13 | 67 (4) |
| **2013** | 350 | 1 | 25 | **324** | 84 (0) | 49 | 172 (19) |

between becoming too distant from the research issue (as this can reduce understanding) or being too involved (which can cloud judgment). We were critically reflexive of our aim throughout the research to understand how diabetes and its epidemic was being talked and thought about in public domain as linked with obesity and the obesity epidemic. In this respect we were open minded about what we what might find and specifically sought out paired methods we could use to enhance the angles from which the data could be considered; one inductive and one deductive. We believed this would support us to maintain empathic neutrality. We report comprehensively and systematically on the findings by year in the findings section before reflecting on the changes across time, to further reflect our neutral approach.

### Techniques of analysis

Studies in this arena have so far used critical discourse analysis [1], content analysis [31], and types of framing analysis [29, 34, 52]. Given the depth and detail pursued in this study, a layered approach to analysis was thought ideal. We decided to combine thematic and framing analysis with the aim of complementing and extending the robustness of knowledge in the field. Thematic analysis was first selected because of its inductive and flexible approach, plus its ability to build familiarity with data and develop rich, detailed descriptions [54, 55].

An innovative type of framing analysis was developed to secondarily analyse data. Framing analysis explores the ways in which issue presentations include or exclude certain information or viewpoints, to promote certain interpretations [56]. It is used often in media research yet only seldom in health and medical circles. Only two studies exploring diabetes used framing principles, applying them through quantitative design to large data samples [34, 52]. A qualitative approach was thought more appropriate here, given the richness and depth of the field of study. We decided to adapt a framing analysis promoted by Matthes and Kohring [57], which used Entman's perspective of a frame being made up of four components, shown diagrammatically in Fig 2 [58]. According to Entman's logic, these frame components make up a frame and all content of a full frame could be attributed to one of these four components.

We reasoned that we could use these frame components to drive coding under pre-decided frames of interest following the thematic analysis as a deductive coding strategy decided a-

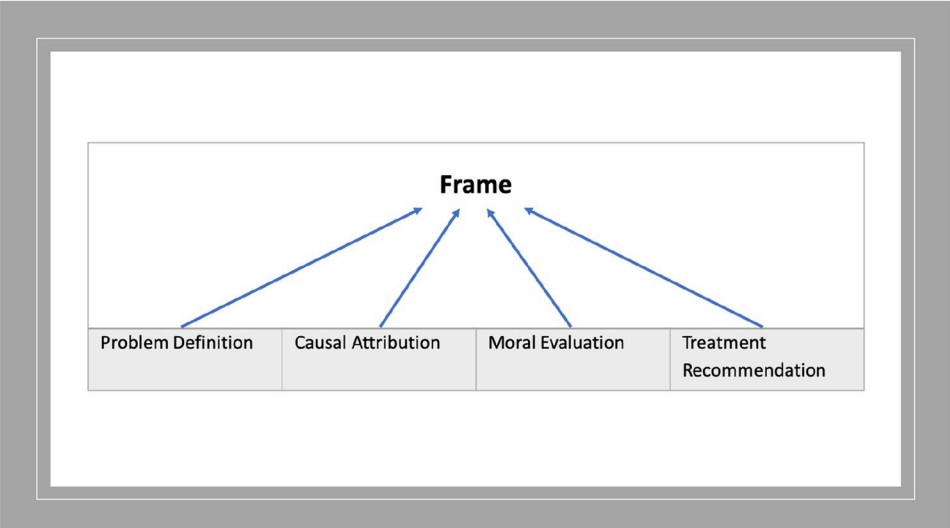

**Fig 2. Entman's perspectives of framing.**

priori. This would provide an opportunity to consolidate and extend findings from the thematic analysis, ensuring there was a systematic approach taken to the data and that findings were triangulated with as much rigour as possible. More information about the method used is published elsewhere [59], and the procedures used for both methods are elaborated below.

## Processes of analysis

**Thematic analysis.** Author 1 started with the sample from 1993, reading the sample to develop familiarity with the themes and content presented [55]. Axial and selective coding following Ezzy [60] in NVivo then developed distinct coding structures within each sample. These were reviewed by Authors 2 and 3, with findings were recorded as 'initial' observations. This process was repeated with 2001, then 2013. At the end of thematic analysis, Author 1 wrote up observations about the ways in which findings differed across the three samples. Authors 2 and 3 reviewed these drafts, providing input for conceptual clarity and rigour.

**Framing analysis.** We conceived that medical, behavioural and societal frames would be useful heuristic devices in this research. These three categories are important to public health where health is conceptualised to exist and operate through medicine, behaviour, and broader society. Findings from prior research in the field could also be organised according to these categories, providing further rationale they would be useful frames to examine within the data.

After the thematic analysis, Author 1 approached the 1993 sample with the a-priori framework (i.e. see Table 3 for an example of the societal frame), re-reading all the data while coding data in NVivo to *only one* of the four frame components under the relevant medical, behavioural, or societal frame (i.e. a total of 12 possible coding locations for each year). Some of the latent meaning embedded in news texts proved difficult to code, and coding concerns were resolved in collaboration with Authors 2 and 3.

The findings from 1993 were recorded as initial observations, and the process was then repeated with the sample from 2001, then 2013. At the end of the framing analysis, Author 1 wrote up observations about the ways in which findings differed across the three samples. At this point, having completed both analyses, Author 1 considered the six groups of initial findings, reading between and across them, and explicating the differences derived from the two methods across the different years. These findings were tabulated and then discussed with Authors 2 and 3, which facilitated refinement of the six categories, enhanced the clarity of how each category differed (or overlapped), and provided working reflections drawn on for later interpretation. No major conceptual changes were made to these categories, however the language used to describe the findings emerged and evolved throughout these discussions.

## Results

Framing of diabetes moved from medical in 1993, to behavioural in 2001, then to societal in 2013. In 1993 (n = 19) there was not a strong focus on links between diabetes and obesity, although obesity was acknowledged as a risk factor for diabetes, and having both diabetes and obesity was portrayed as a risk for more serious medical issues like heart attacks. By 2001 (n = 119) diabetes and obesity were frequently discussed in tandem and obesity was reported as *causal* to diabetes, in addition to being a risk factor. In both 2001 and 2013 (n = 324), increasingly technical gradations of modifiable risk factors for diabetes were highlighted (i.e. diet,

**Table 3. Example of societal coding frame and framing components.**

| Parent Node | Societal Frame | | | |
|---|---|---|---|---|
| Child Nodes | Problem Definition | Causal Attribution | Moral Evaluation | Treatment Recommendation |

exercise and weight). Race, ethnicity and culture however started to be positioned as states of 'high risk' for diabetes during 2001 and these ideas persisted in 2013 articles. Non-modifiable risk factors like age and family history were less prominent than weight, diet and exercise in both 2001 and 2013. Notions of blame and individual responsibility around diabetes emerged during 2001, with a particular focus on pre-/mothering practices. By 2013, concerns around an 'avalanche' or 'epidemic' of diabetes, caused by the prior obesity epidemic, were prominent.

We present our findings in three sections: years 1993, 2001, and 2013. We view that presenting the findings in this order provides the reader with an in-depth understanding of content prominent within each year. Within each section we first present a rubric that details the frames and frame components most prominent in that year (i.e. medical, behavioural or societal) indicated by the shading of the rubric. Darker shades indicate more prominence of that frame component amongst that years' sample. These judgements were made by the researcher based on the quantity and salience of content coded across frame components within each year. In these rubrics we also outline key topics coded to that frame component. Following the rubric we provide subheadings to organise the findings from our thematic analysis, with links made to the framing analysis throughout. The subheadings used change slightly across years to reflect the discursive shifts in how diabetes and obesity were represented within the data.

## 1993

Fig 3 shows the frames and frame components most prominent in 1993.

**Modifiable and non-modifiable risks.** In 1993, rather than diabetes being a main issue of interest in relation to obesity, information about diabetes tended to be communicated alongside or through discussions about heart disease, at that time cited as a major health issue. Diabetes was asserted to be 'part of the ageing process' [61] but dangerous when combined with heart disease and obesity, because it increased the risk of heart attacks.

Obesity was clearly identified as a risk factor for diabetes during 1993.

> . . . obesity, with a BMI of 30 or more. . . is associated with significant health risks, especially heart disease, stroke, diabetes and cancer.

[62]

|  | Problem Definition | Causal Attribution | Moral Evaluation | Treatment Recommendation |
|---|---|---|---|---|
| *Medical* | -Impotence<br>-Joint pain<br>-Death<br>-Heart attack | -Part of the ageing process<br>-Impaired metabolism<br>Risk<br>-Central obesity |  | Management<br>-Keep BGLs down<br>-Pharmaceuticals<br>Prevention<br>-Pharmaceutical 'blockers' |
| *Behavioural* |  | -Sedentary lifestyles<br>Risk<br>-Obesity | -'Diabetics' = 'fatties'/ indulgent | Prevention<br>-Healthier Diet<br>-Exercise<br>-Weight loss |
| *Societal* | -Major cause of absenteeism | -Inactive occupations<br>-Rapid industrialisation |  | -Food Labelling to Support Better Dietary Choices |

**Fig 3. The framing of diabetes in 1993 (n = 19).** In the figure BGL is shorthand for Blood Glucose Levels.

Only one account in this sample identified that weight might be a possible symptom of another disease, rather than a risk [63].

In 1993, the only piece to talk about genetics or ethnicity described the evolutionary mismatch between western diets and the genome of Pima Indians of Arizona. Family history or heritability was not linked to diabetes within this sample.

> They had survived on crops such as sorghum and often faced long periods of hunger until the white man introduced them to fatty foods. Now almost all the elderly members of the tribe are overweight and diabetic because of a genetic tendency to make excess amylin. This is useful when food is scarce as it concentrates glucose on fatty energy reserves. But it can prove disastrous when plentiful fat-rich food is available, as the amylin pours more and more glucose into fat and triggers diabetes.

> [64]

**Interpreting proposed responses to the issue.** Pharmaceutical management was most commonly recommended as a response to diabetes in 1993. The focus on diabetes as a medical problem with medical solutions contributed to diabetes being framed as a medical issue during this time period, despite links made with obesity as a risk factor. Some behavioural conceptions were present, in that sedentary lifestyles were noted to be leading to an increase in chronic disease, including diabetes. Exercise and diet were identified as useful in managing diabetes once it was diagnosed.

One article suggested some moral evaluations of people who had diabetes, although it inconsistently delineated 'dieters' from 'diabetics' throughout the piece, so it wasn't clear if they were inferring that diabetics were big eaters, or that dieters were lucky that they could use the diabetes pill.

> Dieters' dream pill; 'CURE' FOR DIABETES COULD LET THEM EAT CAKE AND STAY SLIM, SAY SCIENTISTS

> [64]

## 2001

Fig 4 shows the frames and frame components most prominent in 2001.

In 2001 medical framings of diabetes highlighted the biological impacts of increased insulin, emphasising its serious, risky and 'deadly' nature. More discrete and nuanced risk categories for diabetes were also outlined. Behavioural framings of diabetes were most prominent in 2001 compared to medical or societal framings of diabetes, and emphasised the role of obesity as a cause/risk for diabetes. In addition, ethnicity/race/culture and lifestyle elements began to be linked with diabetes risk. Societal framings of diabetes started to emerge in 2013, focusing on the increasing rates of diabetes, the consequent scale of diabetes as a national health problem, and the associated financial implications to society.

**Modifiable risks/causes.** By 2001, diabetes and obesity had started to become more topical, evidenced by an increasing number of news articles. Diabetes was often discussed in articles describing the 'obesity epidemic', while other articles focused exclusively on diabetes as an emerging health problem, for which obesity was reported as a risk factor *and/or* a cause.

Discussion about the aetiology of diabetes became more technical, with in-utero risk acknowledged to have some role. During 2001, medical framings emphasised enhanced levels of diabetes risk.

|  | **Problem Definition** | **Causal Attribution** | **Moral Evaluation** | **Treatment Recommendation** |
|---|---|---|---|---|
| *Medical* | -Blindness, amputation, kidney failure<br>-Vulnerability to other chronic illnesses + death | -Metabolic Impairment (BGLs/GT, ins. resistance)<br>-In-utero risk | -Serious<br>-Deadly disease | -Pharmaceuticals<br>-Manage BGLs<br><u>Research</u><br>-Stem cells, islet transplant |
| *Behavioural* | -Disruption of vocation<br>-Low quality of life<br>-Tiredness<br>-Irritability | -Obesity/overweight /fat<br>-Poor Diet<br>-Sedentary Lifestyles<br>-Poor mothering practices | -Mothers = careless about diabetes risk<br>-Lifestyle change = simple<br><br><u>Those with T2DM</u><br>-Are indulgent, lazy<br>-Eat outrageously | <u>Management and Prevention</u><br>-Diet<br>-Weight loss<br>-Exercise<br>-Act on own/child's risk |
| *Societal* | -Cost (health services and national economy)<br>-Strain on health services<br>-Younger ages affected | -Obesity epidemic<br>-Ageing population<br>-Increasing sedentary lifestyles | -Swallowing health budget<br>-Draining economy<br>-Future gen's will suffer<br>-Diabetes = timebomb | -Prevention and Early Detection<br>-Better frameworks to inform these activities |

**Fig 4. The framing of diabetes in 2001 (n = 119).** In the figure BGL is shorthand for Blood Glucose Levels; GT is shorthand for Glucose Tolerance.

All the same, the symptoms are all there: high blood pressure, raised levels of telltale fats called triglycerides found in the blood, and insulin resistance—an acquired resistance to the body's vital glucose-handling hormone. Diabetes and heart disease are lying in wait for anyone with this group of symptoms, collectively known as Syndrome X.

[65]

Behavioural framings of diabetes were most prominent in 2001 in comparison to medical or societal framings. Notably, obesity started to be explicitly labelled as a cause in addition to a risk factor.

One of the main causes of Type 2 diabetes is obesity.

[66]

In addition, being overweight was also implicated as causal to diabetes, expanding the body sizes that were correlated to 'result in' diabetes.

Other consequences of being overweight are more dangerous and include diabetes type 2.

[67]

Sugar was talked about specifically during 2001 as a meaningful part of the weight gain equals diabetes conundrum at this time, extending the equation to: sugar = weight

gain = diabetes. It was more common that sugar was described in relation to risk rather than fats, and as such, sugar become enmeshed in the conceptual picture available to individuals regarding the causation of weight gain *and* diabetes.

> SUGAR: A form of carbohydrate energy, so it's fine in moderation. But too much, particularly spread through the day rather than eaten in just one or two sessions, can lead to tooth decay and can increase the amount of insulin you release, leading, over a lifetime, to a greater chance of weight gain (insulin encourages fat deposition) and diabetes.

> [68]

In 2001, seven accounts noted that age had a role in developing diabetes. This fact was often quoted in order to situate the earlier onset of the disease, particularly in reference to increasing diabetes rates in the UK and most regularly as a partner-in-crime to obesity—the 'trigger' for diabetes.

> Reasons for the predicted increase include an ageing population: diabetes is more common in older people and the epidemic of obesity, which can trigger diabetes.

> [69]

> . . .in 20 years' time more of today's children will be dying from obesity-related diseases, such as type 2 diabetes. . . "We're talking about an epidemic," he says, "and if we don't do something now, the next time we see these children is when they turn up in their thirties with heart disease and diabetes."

> [70]

**Non-modifiable risks/causes.** As with the 1993 sample, only one account provided an alternative account of how weight was related to diabetes, mentioning it as a symptom of Type 1 Diabetes in children [71]. Considering the growth in sample size between these years (i.e. n = 19 in 1993 and n = 119 in 2001), this notion became more marginal. One other 2001 article identified that although better screening may have been driving the increases in Type 2 Diabetes, 'fatness' was the real driver of increased diabetes rates.

> The researchers, from Cardiff University and the Heart of England NHS Trust in Birmingham, said the rise [in diabetes rates] may be partly due to better screening, since GPs now have performance-related pay that rewards them for diagnosing sufferers. However, they said rising obesity was the main driver. Lead researcher Professor Craig Currie said: It s almost entirely obesity. How fat you are is the top and bottom of it.

> [72]

In 2001 four ethnic 'divisions' were referenced in risk for diabetes; with African, Carribean and Asian groups identified to be at five times the risk of other groups [66]. The term 'ethnic minority' was used once in relation to higher risk of the condition [73]. Insulin resistance, identified as a precursor to diabetes, was discussed as particularly common to various 'races' [74].

Despite being a strong indicator for diabetes risk in the extant literature, family history was only associated with diabetes risk in five Daily Mail stories and one from the Guardian. Of those accounts, only one asserted that the risk was correlated with how close the family member was [75]. Another stated personal risk of diabetes as 15% if either one of your parents had

it [66], which under-represents the relationship. Individuals were instead tasked with negating diabetes risk by keeping a healthy weight.

> Q: DO you have a family history of heart disease, diabetes, prostate or bowel cancer? Obesity is a risk factor for all of these conditions, so keeping to a healthy weight as you age can reduce your risk.

> [76]

**Interpreting proposed responses to the problem.** Modifiable risks for diabetes were presented as critical for preventing the condition in 2001. Weight loss through exercise and diet were often promoted as 'solutions' to 'protect' yourself from diabetes risk. Moral evaluations started to surface around people who developed diabetes in light of its preventable nature— neglecting to action 'simple' changes in diet and exercise. In the second quote, a link between individual action and the 'epidemic of Type 2 diabetes' is made.

> This evidence that diabetes is a preventable illness has enormous public health and economic implications. . . anyone with one or more of these risk factors can take effective steps to protect themselves with specific lifestyle changes.

> [66]

> Walking for just 30 minutes a day could halve the risk of developing diabetes, researchers have found. Simple changes in diet and exercise may halt the epidemic of Type 2 diabetes. . . '. . .to go for a walk with your spouse doesn't seem to me to be asking the impossible.'

> [77]

During 2001 it was also identified that obesity and poor nutrition logically 'lead to' diabetes across the entire life span, including in children. The 'poor diet time bomb' is presented as a mix of obesity, sedentariness, poor diet and diabetes in the context of the current generation growing into adulthood.

> POOR DIET TIME BOMB: TYPE TWO DIABETES. . . 'There's a lot of research to show that obesity, lack of exercise and poor diet lead to type two diabetes in adulthood,' says Mairi Benson of Diabetes UK.

> [78]

The notion of children-at-risk was a prominent topic in 2001 and tended to frame the diabetes problem as an individual responsibility issue that had implications for broader society. This coincided with the increased attention on maternal bodies and mothering practices, where in-utero risk, pre-pregnancy diets and early life practices were portrayed as entwined with diabetes risk for children. The attention on mothers' biology (i.e. medical frame) was juxtaposed with a responsibility on mothers to prevent diabetes in children (i.e. behavioural frame).

> In the United States children and young people in their teens are developing this sort of diabetes, and it is linked with obesity. Copelman says that although diet is important, the key factor is exercise. Children are leading couch-potato lives. They do less exercise at school and are driven everywhere because of increasing concerns about safety. Our children's lifestyles reflect ours as parents.

> [79]

The cost of diabetes to society via health services and the national economy, as a result of being overweight, was a dominant topic in the societal frame during 2001. Prevention and early detection were presented as part of the answer to this problem; however this societal issue was often depicted as a result of overweight.

> About 1.4m Britons are diagnosed with diabetes, a figure that will probably double by the end of the decade since more people are expected to become overweight.

[73]

## 2013

Fig 5 shows the frames and frame components most prominent in 1993.

In 2013, diabetes vis a vis obesity was framed as a societal issue much more strongly than as a medical or behavioural issue. An emphasis on 'alarming' rates of the condition carried across from 2001 and 'societal safety' in relation to the epidemic became a prominent theme. Causal attribution (i.e. individual practices) and treatment recommendation (i.e. altering individual practices) of the behavioural frame remained prominent, sustaining the emphasis on diabetes as an issue of individual responsibility—a notion which, when appearing in concert with a societal framing of diabetes, enabled the 'national conscience' to be brought into question, powering up concern around individual practices to an issue of national importance.

**Modifiable risks/causes.**  The ability to position diabetes and obesity as a societal issue during 2013 relied on many of the important developments from prior years. For example, during 2001, obesity and overweight were both described as discrete causes of diabetes. In

| | Problem Definition | Causal Attribution | Moral Evaluation | Treatment Recommendation |
|---|---|---|---|---|
| Medical | -Gateway to poor health<br>-Early death + disability<br>-Impairs body function<br>-Impairs pregnancy/birth | -Metabolic impairment | -Degenerative<br>-Serious<br>-Incurable | -Medication<br>-Control BGLs<br>-Improve insulin sensitivity |
| Behavioural | -Personal cost of illness (amputation + care)<br>-Limited insurance access<br>-Disruption of vocation | -Obesity<br>-Diet<br>-Exercise | -Parents of at-risk children are reckless or lazy<br>-Modifying lifestyle = easy<br>-Those at-risk (fat) = ignorant | Prevention + Management<br>-Diet<br>-Exercise<br>-Weight Loss |
| Societal | -↑ health expenditure<br>-Incidence ↑ + scope of issue not known<br>-Inequitable access to health services<br>-Ignorance of risk factors | -Sugar/modern diet<br>-Obesity<br>-Ageing population | -Crisis/disaster<br>-Public health emergency<br>-Care = shameful<br>-Can't trust food industry<br>-Poor national conscience | -Raise awareness<br>-Improve health services<br>-Improve knowledge of scope of problem |

**Fig 5. The framing of diabetes in 2013 (n = 324).** In the figure BGL is shorthand for Blood Glucose Levels.

2013, weight is used to explain rising rates of diabetes at a societal level more consistently and confidently, continuing the presentation of weight/obesity as causal to diabetes. This was seen across all three papers. No accounts in this sample challenged the view that being overweight could be a symptom of diabetes, rather than a 'cause'.

> ONE in 20 Britons is suffering from diabetes as a result of soaring obesity rates, a shocking study has found.

[80]

> Since 1996 the number of people with diabetes has increased from 1.4 million to 2.9 million, largely linked to weight increase.

[81]

> . . .the academy says doctors are "united in seeing the epidemic of obesity as the greatest public health crisis facing the UK. The consequences of obesity include diabetes, heart disease and cancer and people are dying needlessly from avoidable diseases."

[82]

The salience of sugar in catalysing obesity, and therefore also playing a causal role in the epidemic of diabetes, was portrayed in many articles during 2013. A moral reading on sugar and sugar consumption was also prevalent at this time, and fostered certain interpretations about people seen to be eating sugar.

> Sugar is behind the global explosion in type 2 diabetes, researchers will announce today in a report claiming it plays a uniquely damaging role in a disease that experts fear could overwhelm the NHS. Obesity is usually cited as the main driver of diabetes. But a study by US medical researchers identifies sugar as a predictor of diabetes separate from obesity.

[83]

> Still, it seems ridiculous to liken food to hard drugs—is it an overstatement? There is evidence that sweet foods do hit the pleasure and reward centres in the brain rather like hard drugs. . . leads to insulin resistance (where insulin becomes less effective), high blood pressure, diabetes and unhealthy cholesterol levels—all risk factors for cardiac disease. You don't even have to be overweight for this to occur.

[84]

Further, the danger attributed to diabetes as a medical issue in 2001 was transplanted into 'alarming' reports of diabetes as a societal issue in 2013.

> The charity's Scottish director, Jane-Claire Judson, said: "We are not scaremongering . . . if 18,000 to 19,000 people a year were contracting meningitis or bird flu we would be really, really concerned. It is a major problem."

[85]

**Non-modifiable risks/causes.** Only 14 of the 324 articles in 2013 described ageing having a role in diabetes development, or an ageing population influencing diabetes statistics. Unhealthy lifestyles and fatness were at times used to minimise the importance of ageing in

the causal profile of diabetes. At other times, they were presented as the two key drivers of diabetes in the UK.

> The soaring rate of type 2 " which mostly affects middle-aged people " is being blamed on obesity.

[80]

> With the number of diabetics expected to double from the present 2.5 million to 5 million within a generation—a consequence of an ageing population and unhealthy lifestyles—they say that the NHS risks being overwhelmed by the cost of treatment.

[86]

In 2013, 13 articles identified family history in T2DM aetiology: 11 from *the Daily Mail* and two from *the Times*. Similarly to 2001, however, the importance of familial risk in disease aetiology was understated in relation to other risk factors, most notably weight *and/or* 'lifestyle', often used loosely to refer to being overweight.

> Some are genetically more at risk but lifestyle is the key trigger, he said, adding: It's still quite rare but we do see them. In the huge majority of cases type 2 diabetes develops as a consequence of being overweight. So as the weight of the nation increases, the incidence goes up too.

[72]

> "Certain genes are dictators, like the genes for blue eyes or brown hair. There's nothing you can do about them. The genes for Alzheimer's and other diseases such as diabetes are more like committees: they make suggestions. Lifestyle factors have a profound effect on determining our health, regardless of our family's medical history. What we eat and drink can help determine who thinks clearly decade after decade and who is affected by a terrible tragedy."

[87]

Only one article in 2013 endorsed the importance of a genetic influence in the development of diabetes, in addition to weight and lifestyle.

> Type 2 diabetes is normally associated with obesity and a couch potato lifestyle—not heroic athleticism. Even now no one can tell for sure why he developed it—but it could be that it was unlucky genes. . . Although the biggest risk factor for type 2 diabetes is weight, genes can also have an important role. . . If one parent has type 2, then you have a 30 per cent chance of developing it. If both parents do, then that risk is 50 per cent. It is possible to develop type 2 diabetes even if you are of a normal weight—just because of the genetics you inherited.'

[88]

Ethnicity also became an identifier in evaluating diabetes risk during 2013. 11 accounts discussed links between diabetes and ethnicity, tying it in with links to lifestyle and weight. Eight articles talked exclusively about South Asians, two also refer to black, Caribbean or African groups, and one article only uses the term 'ethnic'. These groups were established to be at

higher risk at a younger age and lower level of obesity—maintaining the link between diabetes and obesity [89, 90]. The aerobic fitness of these groups was presented to be lower than other ethnicities [91], and specific gyms were marketed as important for young Asian females at high risk of the disease [92]. A testimonial account unpacked the experience of living at such high risk for the disease:

> 'I didn't want to be skinny, I wanted to be fit and strong, to have the energy to play with my kids, look good in my clothes and reduce my risk of problems like heart disease and diabetes. As an Asian woman, my ethnicity is proven to put me at greater risk of developing them.'
>
> [93]

Two stories released on the same day in *the Times* and *the Daily Mail* reported on the high levels of diabetes in some boroughs of London. *The Daily Mail* piece extended the links genetic risk had with 'race' to culture and lifestyle, and the interplay of these elements.

> Almost one in five people in Brent is from India: diabetes is especially prevalent among people of South Asian origin.
>
> [89]

> Scientists believe this is partly down to a genetic difference which means that their muscles do not burn fat well. Traditional cooking methods using oil, creamed coconut milk and high-fat butter called ghee may also play a role.
>
> [94]

**Interpreting proposes responses to the problem.** Critiques of the necessary public health response in news articles mirrored the notion that diabetes was caused by unhealthy lifestyles, for which obesity was often a symbol. The recommended response tended to be broad scale prevention of diabetes via awareness raising and education, particularly around the role of lifestyle and weight. While a 'pandemic of inactivity' was described to be contributing to society's diabetes problem, the inferred responsibility for solving this problem circulates around individual action rather than societal change.

> The charity's chief executive Barbara Young said the aim is to 'lay to rest the myth' that type 2 diabetes is a mild condition. 'This is a misconception that is wrecking lives and is the reason that as a country we are sleepwalking towards a public health disaster of an almost unimaginable scale,' she said. Losing weight, eating more fruit and vegetables and becoming more active are thought to cut the risks of diabetes.
>
> [95]

> Four out of ten men and five out of ten women are still not active enough to benefit their health. This increases the risk of serious illnesses like type 2 diabetes, heart disease and certain cancers, and makes it more likely that people will be overweight or obese. . . [this] brings us closer to understanding the kind of societal shift that needs to happen before we truly combat the pandemic of inactivity. The figures are alarming and show that we need to take action now.
>
> [96]

Another excerpt from the same Times piece denotes that this 'pandemic' is a result of non-compliance from medical advice, rather than an unintended impact of structural forces such as industrialisation.

Thousands of cases of cancer, heart disease and diabetes would also be prevented if people were as active as doctors recommend. . . The death and illness toll of inactivity is comparable to that of smoking, they say.

In 2013, the problem/s of diabetes and obesity also started to be portrayed as issues of societal safety. Warnings suggested that together they posed significant threats to societal and economic progress.

Poor lifestyles, increasing sedentary habits, obesity and diabetes threatened to eradicate advances made in Britain in recent years . . .

[97]

Soaring levels of obesity, which have been predicted to double by 2030, coupled with rising levels of diabetes and heart disease linked to poor diets and a lack of exercise, will overwhelm the dwindling numbers of nurses.

[98]

The children-at-risk narrative which was visible in 2001 was also drawn on to further demarcate diabetes as an issue of public safety by 2013. Although mothers still received specific attention in 2013 news articles, the focus broadened to include 'parents' as responsible for diabetes prevention, perhaps in line with reporting that paternal sperm could be attributed to diabetes risk. These accounts are much more definitive in the tone and extent of their accusation than in the previous years, and link obesity or poor nutrition with diabetes.

We're used to hearing words like "obesity" and "diabetes" and "risk". We're also used to switching off. But this, unfortunately, is because we don't think about what it really means. . . we Brits are the biggest in western Europe. This is not a joke. This is, in fact, the opposite of a joke. We know what makes people fat. It's bad diet and not enough exercise, not cooking properly and not eating enough vegetables, and drinking too many fizzy drinks. And it's growing up with parents who don't care enough about what you eat.

[99]

Are we destined to self-destruct? Not necessarily. It seems particularly terrible to me that small children are developing type 2 diabetes because of what their parents feed them.

[100]

## Missing features of diabetes presentations across the years

Notably, while poverty has long been associated with diabetes this idea was not well-represented in news articles from any year. In 1993, no connection was made between poverty, social context and diabetes. In 2001, two narratives did engage with the inequitable distribution of diabetes by income distribution; however they were connected through the types of lifestyles lower socioeconomic groups had, rather than the types of lifestyles that their resources afforded or encouraged them to take up.

Dr Ian Banks, president of the Men's Health Forum, said that a person's life expectancy almost mirrored their social deprivation. He said people in low income brackets tended to smoke more and eat less healthily than professionals. They tended to suffer higher rates of obesity, cardiovascular disease and diabetes than those in higher income brackets. "They are getting the killers that reduce life expectancy," he said. The gap between the richer and poorer in Britain was growing.

[101]

In 2013, there were a number of equity narratives outlining geographical variations in diabetes care. The only piece in 2013 to connect poverty, processed diets and diabetes risk was a sessional piece published in the Guardian. This article also challenged the idea that diabetes was an issue of individual responsibility. However, they advocated for stricter nutritional guidelines amongst the food industry, rather than targeting upstream issues.

Burger jokes have been ten a penny over the past week, but the impact that this sort of junk has on people's lives is not funny at all. People on low incomes suffer far higher rates of diet-related disease, and not just obesity. . . Mothers from low-income groups are more likely to have children of low birthweight, who, in turn, are likely to suffer poor health and educational prospects as a result. They have more childhood eczema and asthma. They have higher rates of raised blood pressure, thanks to their processed diets. They are more likely to suffer diabetes, heart disease, vascular disease and strokes. . . Adulteration of food, legal or otherwise, is no laughing or sneering matter.

[102]

## Variations between newspapers

There were some variations in the reporting styles and tendencies of newspapers which should be noted to provide a deeper context through which to interpret the findings. Firstly, *the Daily Mail* had more health articles than either of the other papers, and also tended to run more sensationalist reviews of health issues than the other papers. *The Times* and *Daily Mail* often reported on the same content on the same day, however *The Guardian* also used the same copy at times indicating their link to the Associated Press from which they gather daily content.

*The Times* had a more economic focus than the other two papers and often highlighted the economic issues associated with the problem of diabetes. *The Guardian*, despite being politically centre-left, exhibited many of the same trends towards obesity as 'causal' to diabetes. While moralising judgments about the development of diabetes were less common they were still present. Considering *the Guardian's* assertions of independence and advocacy, perhaps it could point to a relatively pervasive uptake in public discourse of the assumptionist perspectives highlighted in this study.

Further, throughout the analysis it emerged that the mentions of obesity as 'causal' to diabetes were present across many different types of articles. During coding articles were categorised into those reporting on medical research, health intervention, 'lifestyle' advice and opinion pieces, and all of these mentioned obesity as a cause for diabetes. Significantly, the near-total absence of counter-narratives shows that the dominant meanings around diabetes in conjunction with other key terms were remarkably widespread.

## Discussion

This discussion aims to provide a brief description of changes in diabetes presentations which occurred over the time period that the obesity epidemic was gaining traction in public discourse. While possibilities for 'civic-oriented journalism' may be drawn from the following discussion [103], the intention remains on exploring the way in which socio-cultural discourses appear to be reflecting nuanced discourses about health, risk, illness, blame, and responsibility.

### Type 2 diabetes (and obesity): The collapse of risk into 'causality'

*"It doesn't seem surprising to learn that obese children are more likely to have. . . signs of the early stages of type 2 diabetes—a disease traditionally associated with overweight adults"*

[104]

Between 1993 and 2001, obesity transitioned from being portrayed as a 'risk' factor for diabetes to a 'cause', a change that was naturalised by 2013. This finding coincides with other suggestions that diabetes causality is described with deceptive simplicity in western media [1, 29, 31, 34, 52]. It also further legitimises concerns that obesity and diabetes are being convincingly conflated in public literature [1, 32, 33]. The narrowing of diabetes presentations between 1993 and 2001 provides reasonable evidence this convergence was influenced by the WHO [38] Obesity Report and associated public discussion.

It is likely that epidemiological evidence has been playing a role in strengthening perceived associations between factors which are not necessarily causal [10, 13, 105, 106]. Shortly after the WHO report was released (i.e. 2000), links started to be made between obesity and diabetes in the same populations [1]. England are specifically mentioned in the WHO report as having doubled their 'obesity levels' in a relatively short period of time [38], which may have encouraged specific measurements of these factors in the UK. The report also raises that even *'relatively small increases in body weight'* influence health risks [38], perhaps contributing to the prominence of overweight as well as obesity in discussions about health risks.

The sheer physicality of fatness is difficult to hide [107]. The larger body can provide a very strategic reference point for a neoliberal governance enacted through visibility and surveillance [108], often as a symbol of an 'unhealthy lifestyle' as evidenced by the interchange of these terms in 2001 and 2013. Other visible features of diabetes risk, however, also became prominent between 1993–2013 –even in a sample focused on exploring linkages between obesity and diabetes. Race, genetics, ethnicity and culture were arenas in which diabetes risk was communicated, particularly so in 2013. The construction of 'ethnic groups' existing at such 'high risk' for diabetes opens up opportunities to legitimise concern about particular racial groups through health discourse [3, 109]. Patterning diabetes risk to visible characteristics is a key cause of suspicion in how health discourses are generated in the new public health.

As Fee [110] noted, tying diabetes (and obesity) with ethnicity enables obfuscation of the fact that so-called 'ethnic groups' who experience higher rates of diabetes are also often significantly disadvantaged, which may actually be the cause for their increased risk of diabetes and/ or obesity. Aspects of nutritional discourse in this way present blank slates onto which ideals can be projected in contemporary industrialised life [111]. Keval [112] argues that the term 'ethnicity' in the UK is widely being linked to notions of 'disease' which centre around culture and lifestyle, particularly for South Asian populations. This may consolidate the illusion that ethnicity is really 'the problem' rather than the way society is unequally structured. Ongoing

academic work into ethnic groupings of diabetes [113] must ensure that effects of poverty and marginalisation are not repackaged as 'genetic flaws'.

## Type 2 diabetes (and obesity): The expansion of risk

*"Type 2 diabetes is something we could all develop."*

[85]

By 2001, diabetes risk was portrayed as ubiquitous: everyone, everywhere, could be or become at-risk of the disease (in conjunction with obesity and overweight). Understanding that one's health is always in jeopardy [3] is acknowledged to keep one attentive to the possibility of future illness; effectively what was once 'normal' becomes 'problematised' [114, 115]. Campos [12] identified that discourses around obesity were significant because they enabled what was initially normal to be viewed as problematic, because everyone everywhere is at risk of becoming too fat. The evolution of diabetes presentations in the news, vis a vis obesity, showed similar curvature.

Through linking diabetes with obesity, the importance of preventing weight gain is further reinforced. The fact that diabetes risk is entwined with food and consumption ensures that a wide range of necessary nutritional practices stay problematized. During 2001 and 2013, the enhanced specificity and complexity of diabetes prevention mechanisms focused around food further imbued a range of food practices with convincing concern about diabetes risk. This creates a range of new sites of guilt, blame, risk, and stigma [116]. By 2013, sugar was thought to confer such a high degree of risk for the condition that it was likened to hard drugs.

This shift enhances the attentiveness required to eating, an everyday activity essential for human survival—both for one's own safety, for potential offspring and for broader society. The attentiveness required is bolstered by the increased hazardousness and alarm with which diabetes was presented. For those who do not have the resources to access a so-called healthy diet, this is likely to engender anxiety, guilt and stress; particularly around the emotionally-charged issue of children's risk. The emergence of formalised, technical gradations of diabetes risk during 2001 and 2013 contributed to outlining the wider range and degree of risk potential that could be applied to the normal population. As Hacking asserts [105], these epidemiological categories are foundational to bringing risk into the everyday space. All of these shifts foster an enhanced dependence on the practice of medicine as a whole [108].

## Type 2 diabetes (and obesity): An evocative epidemic

*Epidemic*: NOUN

1. "A widespread occurrence of an infectious disease in a community at a particular time."

2. "A sudden, widespread occurrence of an undesirable phenomenon."

*Synonyms*: outbreak, plague, scourge, infestation

[117]

The convergence in diabetes and obesity presentations at an aetiological level appears to have enabled the introduction of the diabetes epidemic as the twin or result of the obesity epidemic, as McNaughton [1] earlier suggested. After the issue of diabetes was constructed as an issue of personal failure in relation to obesity, this logic could then be extrapolated to understand

diabetes at a societal level, portraying certain (larger-bodied) groups as having failed their duties of citizenship. The evocative imagery of epidemics are disturbing and unethical [7]. The assertion that diabetes is obesity's 'twin epidemic' appears to be facilitating the furthering of moral panic in health discourse [7, 118–120], first described by Cohen [120] as a legitimising process of public scorn, suspicion, and fear, where the causes of societal problems are blamed on certain groups. He called these groups folk devils.

As in other work [118, 121] mothers tended to be constructed as the folk devils of moral panic around diabetes (although parents do also play a more minor role in these discourses). The introduction of children's safety into diabetes discussions moved this issue outside of public health into broader societal critique, advancing moral panic into new territory [107]. Ulijaszek and McLennan [36] in their review of obesity policy documents also identified that risk and danger debates pointed to caregivers as failing their 'moral' responsibilities and citizenship, while the societal inequalities that led to those risks and dangers remained invisible. Similarly, it also appears that certain 'ethnic groups' are being constructed as likely to develop diabetes [110, 112, 122], mixed with assertions of 'unhealthy lifestyles' with disregard to the social inequalities that are probable to underpin diabetes risk. Claims that diabetes could 'destroy' or 'reverse' Britain's progress demand individual responsibility for diabetes (and obesity) prevention, and are troubling because the perceived blame for such grave issues rests on societal groups who already experience disadvantage.

The capacity of the 'obesity epidemic' to maintain such a stranglehold on the political economy of risk and individual action is thought possible because of entrenched moral and cultural ideologies about fatness [37, 123, 124]. The 'obesity epidemic' was already being understood in the UK as an issue of over-indulgence [36], so it is likely that a diabetes epidemic regenerates moral concerns about food, the body and fatness in the same way which health discourse around obesity has done [107]. Reductive discourses like these undermine human wellness through the sanction of stigma and marginalisation [125], and often work in direct opposition to the human rights principles that underwrite contemporary public health [4].

## Strengths and limitations

Not having a full-blown tabloid (i.e. Daily Express, Star, Mirror) in our sample may have been a small limitation, however they were not fully indexed at the time so not accessible for research purposes. Several tabloids do however draw their content from the same newsgathering organisation (Associated Press) as those included, so there was likely to be some overlap in the content of reports [50]. Our reasoning in selecting 1993, 2001 and 2013 as sample years was based on our logic around key events related to the obesity epidemic, which we cannot claim to be representative of total shifts in diabetes presentations during this time. The paired thematic and framing analysis was a new innovation and as such required ongoing reflexivity around how principles of thematic and framing analyses could or should be translated into our analysis. We feel however this showcases flexibility of research design according to social context, consistent with good quality qualitative research [126]. Overall, our study adds usefully to the critical literature around postmodern health epidemics and provides the first longitudinal analysis of the emergence of the diabetes epidemic in relation to the earlier obesity epidemic in UK news material.

## Conclusion

A coalition of moral, cultural and political discourses about health and lifestyle appear to be at play in shaping how diabetes and its epidemic are represented and understood in UK media in conjunction with obesity. Shortly following announcements of the 'obesity epidemic', obesity

became more central to descriptions of diabetes. This included the shift from obesity as a risk factor for diabetes to a cause of the disease, as well as the 'diabetes epidemic' being portrayed as a direct consequence of the 'obesity epidemic' as has also been shown in Australia and Canada [1]. Together, these events likely extend and consolidate medico-moral concerns about the fat body [7] and the 'fat-tainting' of contemporary health more generally [9].

The emergence of 'moral panic' is evident in relation to a 'diabetes epidemic' and appears to be driving and reinforcing structural inequalities, by enabling some groups to re-create their success through the display of the generative 'healthy lifestyle' concept in a healthist society [6]. This can promote blame-oriented tendencies [127] towards those who appear to have or inhabit 'unhealthy lives', often a consequence of social inequality. The emergence of such certainty about the individual role in bringing on diabetes demonstrates the discursive possibilities which surround health, risk and 'epidemics', but also the adaptive possibilities of neoliberal governance. The freeform adoption and evocative imagery of the 'epidemic' in relation to diseases linked with individual responsibility is disturbing and unethical [7]. The use of reductive and responsibilising discourses have been shown to undermine human wellness [125] through sanctioning stigma and working in direct opposition to the human rights principles that are supposed to underwrite contemporary public health [4]. While the 'new' public health has already departed from its original vision to reduce health inequities [128], a more critical public health must continue to expose the pervasive and adaptive nature of these hegemonic discourses.

## Author Contributions

**Conceptualization:** Kristen Foley, Darlene McNaughton, Paul Ward.

**Data curation:** Kristen Foley, Darlene McNaughton.

**Formal analysis:** Kristen Foley.

**Investigation:** Kristen Foley, Paul Ward.

**Methodology:** Kristen Foley, Darlene McNaughton, Paul Ward.

**Supervision:** Kristen Foley, Darlene McNaughton, Paul Ward.

**Writing – original draft:** Kristen Foley.

**Writing – review & editing:** Kristen Foley, Darlene McNaughton, Paul Ward.

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
