## [Decision Letter · Decision Letter 0]

12 Sep 2019

PONE-D-19-22797

Monitoring the ‘Diabetes Epidemic’: A Framing Analysis of UK Print News 1993-2013

PLOS ONE

Dear Mrs Foley,

Thank you for submitting your manuscript to PLOS ONE. After careful consideration, we feel that it has merit but does not fully meet PLOS ONE’s publication criteria as it currently stands. Therefore, we invite you to submit a revised version of the manuscript that addresses the points raised during the review process.

Thank you for the opportunity to review this critical, sociological analysis of UK media coverage of the so-called 'obesity epidemic.' Please respond to each of the reviewers' comments below and particularly, note that reviewer #1 has submitted their comments in the attached file.

Qualitative, critical approaches are still relatively new to many of PLoS One's reviewers and readers. I encourage you to carefully consider each of the reviewers' comments, but you may wish to provide some explanation and justfication of your critical approach both in the response and in the paper, in lieu of making the requested changes; for example, justifying the use of the term "fatness,"  explaining why it is generally inappropriate to quantify the proportion of articles using each frame, or the purposive (rather than "arbitrary") nature of sampling.

We would appreciate receiving your revised manuscript by Oct 27 2019 11:59PM. To enhance the reproducibility of your results, we recommend that if applicable you deposit your laboratory protocols in protocols.io, where a protocol can be assigned its own identifier (DOI) such that it can be cited independently in the future. For instructions see: http://journals.plos.org/plosone/s/submission-guidelines#loc-laboratory-protocols

We look forward to receiving your revised manuscript.

Kind regards,

Quinn Grundy, PhD, RN

Academic Editor

PLOS ONE

Journal Requirements:

2. In order to meet PLOS ONE's reporting requirements we would be grateful if you could ensure that you have included the following information in your manuscript:

A) Detailed inclusion and exclusion criteria for articles included in your study, in sufficient detail such that another researcher could replication your study.

B) Indicate the number of articles excluded from your study in Figure 1, with reasons.

C) Indicate the number of articles included from 2013 in Figure 1 (current figure indicates "N = XX").

D) Update the Methods section of your manuscript to indicate the full search strategy and keywords used.

Reviewers' comments:

Reviewer's Responses to Questions

**Comments to the Author**

1. Is the manuscript technically sound, and do the data support the conclusions?

Reviewer #1: Partly

Reviewer #2: Partly

2. Has the statistical analysis been performed appropriately and rigorously? 

Reviewer #1: N/A

Reviewer #2: N/A

3. Have the authors made all data underlying the findings in their manuscript fully available?

Reviewer #1: Yes

Reviewer #2: Yes

4. Is the manuscript presented in an intelligible fashion and written in standard English?

Reviewer #1: Yes

Reviewer #2: Yes

5. Review Comments to the Author

Reviewer #1: To be able to judge the manuscript as technically sound more detail is needed in the methods section, particularly around the search strategy and data screening process, as well as more information about the process of analysis. I have provided specific comments on these aspects in the attached file.

Reviewer #2: The paper investigates how the issue of diabetes was represented in UK newspapers overtime. While the topic is important as diabetes has reached the epidemic level in UK, I have several concerns regarding the methodological appropriateness and clarity of the paper.

1. Regarding the sampling of newspaper articles, authors might want to further justify why three specific years were chosen. it seems very arbitrary why 1993, 2001 and 2013 were selected and there is no evidence provided whether the news stories in these three years are representative at all in examining the shifts in the diabetes discourse.

2. It is also not clear how thematic analysis and framing analysis are different from each other. Given the vagueness around the concept "frame" and "framing," it is important that authors carefully review past literature to narrow down the concept to justify framing analysis as a content analysis method, especially when combining it with other methods.

3. As the focus is on the changes in how diabetes is depicted in news discourse, the authors might want to also provide descriptive data quantifying the proportion of articles using each frame for each selected year under analysis before going into details regarding the specific elements included in the frames? It will be easier for readers to interpret the results. Also, the result section seems to be loosely structured which makes it a bit hard to follow.

6. PLOS authors have the option to publish the peer review history of their article (what does this mean?). If published, this will include your full peer review and any attached files.

Reviewer #1: No

Reviewer #2: No

---

## [Author Response · Author response to Decision Letter 0]

23 Oct 2019

Editor Response

Thank you for the opportunity to review this critical, sociological analysis of UK media coverage of the so-called 'obesity epidemic.' Please respond to each of the reviewers' comments below and particularly, note that reviewer #1 has submitted their comments in the attached file.

Qualitative, critical approaches are still relatively new to many of PLoS One's reviewers and readers. I encourage you to carefully consider each of the reviewers' comments, but you may wish to provide some explanation and justfication of your critical approach both in the response and in the paper, in lieu of making the requested changes; for example, justifying the use of the term "fatness," explaining why it is generally inappropriate to quantify the proportion of articles using each frame, or the purposive (rather than "arbitrary") nature of sampling.

 Thank you for the invitation to respond to the reviews provided, and for the helpfulness of the reviews, particularly so #1. This has helped us to improve the paper. In the table below I have outlined our responses to each point, whether or not this be to revise the paper or strengthen our rationale for particular decisions questioned by the reviewer/s. 

In revising the manuscript, we have also considered the newness of critical qual approaches to PLoS One’s audience and attempted to provide further transparency where possible. We have also edited some areas to keep the paper as concise as possible. 

 I have double-checked that our manuscript meets these style requirements. 

2. In order to meet PLOS ONE's reporting requirements we would be grateful if you could ensure that you have included the following information in your manuscript:

A) Detailed inclusion and exclusion criteria for articles included in your study, in sufficient detail such that another researcher could replication your study.

B) Indicate the number of articles excluded from your study in Figure 1, with reasons.

C) Indicate the number of articles included from 2013 in Figure 1 (current figure indicates "N = XX").

D) Update the Methods section of your manuscript to indicate the full search strategy and keywords used.

 A) We have provided more detailed information about this, to enable replication by another author. 

B) I have amended this Figure and included a table which describes the exclusion processes in more depth. 

C) I have revised the provision of this information to a figure of the search terms and a table with more detailed exclusion/inclusion. Apologies for the omission. 

D) We have updated the methods section to indicate the full search strategy and keywords used. 

Please note Reviewer 2 is included first then followed by Reviewer 1 

Reviewer #2 Response

The paper investigates how the issue of diabetes was represented in UK newspapers overtime. While the topic is important as diabetes has reached the epidemic level in UK, I have several concerns regarding the methodological appropriateness and clarity of the paper.

1. Regarding the sampling of newspaper articles, authors might want to further justify why three specific years were chosen. it seems very arbitrary why 1993, 2001 and 2013 were selected and there is no evidence provided whether the news stories in these three years are representative at all in examining the shifts in the diabetes discourse.

 I have made it clearer in the materials and methods section why we purposively sampled from years of 1993, 2001 and 2013. I have also revisited these ideas in the limitation section and identified we cannot claim for this knowledge to be representative of total shifts in diabetes discourse over time. 

2. It is also not clear how thematic analysis and framing analysis are different from each other. Given the vagueness around the concept "frame" and "framing," it is important that authors carefully review past literature to narrow down the concept to justify framing analysis as a content analysis method, especially when combining it with other methods.

 We have provided more detail about the framing analyses used by others in the field and our use of a framing concept for a-priori for deductive coding. Since this paper was drafted we have published a paper in Qualitative Health Research which describes the methodology for this paper in more detail, and provides a review of framing analysis in research disciplines over time. This has been indicated in the paper so readers can pursue further, specific information regarding the development of the method, as they require. 

3. As the focus is on the changes in how diabetes is depicted in news discourse, the authors might want to also provide descriptive data quantifying the proportion of articles using each frame for each selected year under analysis before going into details regarding the specific elements included in the frames? It will be easier for readers to interpret the results. Also, the result section seems to be loosely structured which makes it a bit hard to follow.

 I have made it clearer in the introduction to the results section how the frames are represented throughout the years – this makes the detail regarding specific elements easier to digest, and the rest of the results section easier to follow. Given that we innovated a new framing analysis, which differs from content analysis and did not code whole articles to a frame, it is not possible to quantify the frames present in this way. 

I have also revised the introduction for each years’ section, to more clearly highlight the frames most prominent within each year, before going on to provide the detail in the body of those subsections. 

Reviewer #1 Response 

To be able to judge the manuscript as technically sound more detail is needed in the methods section, particularly around the search strategy and data screening process, as well as more information about the process of analysis. I have provided specific comments on these aspects in the attached file. 

Additional File Follows 

This paper reports on a longitudinal analysis of how the ‘diabetes epidemic’ dialogue has played out in UK newspapers over a 20-year period. While this is an interesting and novel study, which has been well situated within the literature, there are some concerns around the methods and the results that I believe need to be addressed prior to publication. 

Introduction 

Main text, pages 5-6: You have touched on the issue that diabetes is a complex disease for which we do not know the exact causes, but, given the focus of your analysis on how the causes/risk factors for diabetes are presented, I think it would strengthen your manuscript to include a more detailed discussion of how the evidence base and our understanding of the factors associated with diabetes have shifted over the 20 years that you selected for your sample – this would allow for a more nuanced understanding of the shifts in discourse – for example, to what extent do some of the findings represent new understandings about diabetes (I noted for example that a number of your quotes included experts commenting on diabetes)?

 While we recognise this may be strengthen the article, sufficiently tracing the archaeology and complexity of diabetes evidence between 1993-2013 is worthy of another publication on its own and therefore outside the scope of this paper. Further, given the feedback provided later around being more explicit about the relationship between diabetes and obesity (as drawn from our search terms), the key events in this story are reflected in our introductory sections. 

Page 3, line 63: “…constructed in public imagination” – I suggest changing to “constructed in the media” or similar given that the focus of your analysis is only on the content of news media and not how this is interpreted by publics (construction of issues within the ‘public imagination’ will be the result of an interplay between the content of the message and how it is received)

 Thank you for this point, this has been adapted.

Page 4, line 86: What is meant by the term ‘Super Value’ in reference to health? This needs explanation for those not familiar with the term

 I have removed this term and instead explained its implications, rather than using the term without implications. 

Page 5, line 105: “Some authors point out that some evidence… was overstated” – can you be more specific here about what evidence is thought to have been overstated?

 I have included this information, regarding how obesity was measured and classified. 

Page 7, line 162: Here and in other places throughout the manuscript you have used the term ‘fatness’ instead of obesity, and these terms seem to be used interchangeably throughout. Is there a particular reason for this word choice? If not I would suggest simply using the more widely used (and less ‘loaded’) term ‘obesity’ throughout

 The term fatness is often used in the critical health sciences to resist the ‘medicalisation’ of fatness – a physical bodily state for which health risks have potentially been overstated. This is well-represented in critical work around obesity (i.e. Gard and Wright; Boero). I have included a sentence early in the paper which contextualises and justifies the use of this term. 

Materials and methods

Page 8, line 185: “databases – name them – from the early 1990’s” – I assume that the names of the databases should have been inserted here? Yes – apologies. This has been amended. 

Page 8, line 188: Stated that The Sun newspaper was included, but Table 1 lists the Daily Mail – please clarify which of these was included in the sample. Also stated that ‘not having a tabloid’ was a limitation – The Sun is considered to be a tabloid in the UK, as is the Daily Mail (as you state in Table 1 where you refer to it as mid-market tabloid)

 Yes – apologies. This has been amended. I have changed the language to talking about not having a ‘full’ tabloid as a limitation, and moved it to a limitations section between the discussion and conclusion. 

Page 8, Line 191: Can you clarify which papers in your sample draw from Associated Press to see how much overlap there is likely to be within your included sample, as well as the overlap with the papers that weren’t included

 All three of the newspapers we include draw at least some of their content from Associated Press, although there are no publicly available figures regarding to what extent this occurs. In the discussion section we comment on some of the crossover between content in all 3 papers in particular time periods. 

Page 9: There is insufficient discussion of how the combinations of keywords were selected. In particular, I note from the Figure that at no point in the search strategy was the term ‘diabetes’ entered on it’s own, without being linked with obesity and/or epidemic. This is a potential problem when it comes to discussing the number of papers discussing diabetes as an epidemic or in relation to obesity in your sample, as you don’t have the sample of papers that talk about diabetes in the absence of these terms, which would be your denominator

 Thank you for this point – this is a critical translation point for our readers. We have made it clearer throughout the paper that we intentionally searched diabetes vis a vis obesity and made sure in the results and discussion to present our knowledge about the diabetes epidemic in reference to the obesity epidemic, rather than speaking broadly about diabetes. 

Figure 1 Search Strategy: Missing number of papers identified in 2013, and needs to state which search fields were used – title, full text, excerpt?

 Thank you for this detail. I have adapted the figure and included that the search fields used were full text. 

Page 9: At what point in 2013 was the search performed? Do you have a complete 12 months of data for this year? This is important in considering how the number of articles changed across the years.

 The search was performed early 2014. I have added this information at the start of the methods section so it is clearer the full calendar year would have been searched. 

Page 9: Important information on the data handling process appears to be missing:

• How did you get from the 4280 articles originally identified to the 51, 119 and 324 papers across your 3 sample years? 

• How many articles were identified in each database? How did you deal with duplicates across Factiva and Lexis Nexis (and why did you search in both databases if they both contain the same newspapers?)? 

• Was the selection of articles for each of the years done automatically (e.g. by selecting an option in the database just to download articles from those years), or manually by going through and selecting all from those years? 

• Was there a screening process to include/exclude papers according to a set of criteria? If so what were these criteria, did you review the full article text or just an excerpt, who did the screening and what reliability processes were in place (e.g. second person checking a subsample)? 

• Were all records imported into NVivo and handled exclusively in NVivo or were there other data handling procedures?

 Apologies for these important omissions. I have now included information regarding: 

• The process of moving from 4280 original articles to the final number for each subsample

• The numbers of articles identified in each database, how duplicates were removed, and why both Factiva and Lexis Nexis were used 

• Selection was completed automatically – I have added this detail

• I have included these details and highlighted the role of authors 2-3 in these processes

• I have outlined the data handling processes more explicitly from the searching databases to a reference management database and then to NVivo. 

Page 10, line 225: Stated that studies in this area have used thematic, CD and framing analysis – please provide references to key studies here for referecen

 I have detailed these details as suggested.

Page 10, line 227: Stated that “Using a method different to those used so far was thought to extent and complement…” but then you go on to say you used thematic and framing analysis – both of which you previously stated are common approaches in this area. Please clarify your meaning here.

 Thank you for this point. I have clarified the value of combining the methods. 

Page 11, lines 241-243: Is the statement about piloting of computerised coding frame relevant to your methods? It seems unnecessary and I would suggest cutting for brevity

 I have removed this sentence so as not to confuse the reader. 

Page 12 Lines 255-261: More detail needed around the process of thematic analysis. Was this performed just by one author (and which one)? Was there any discussion process with other researchers to refine/explore interpretations? Was there an iterative process to coding?

 I have made the roles of each author clearer and made the processes of refining and exploring interpretations explicit. 

Page 12, lines 263 – 267: More detail needed around the framing analysis: 

• How did the researchers come up with/decide on the three frames? Was previous research consulted here? To what extent did these emerge from the inductive process? 

• Need to define the various frames (the four from Entman and the three identified here) and provide examples to make it clear how these were applied (this could be in supplementary materials if needed). 

• Who performed the framing analysis? Just one author? Did a second person check for consistency?

• The three frame categories used are important analytical and conceptual devices within public health. I have made this reasoning for their use clearer within this section, and also commented on how these three frames aligned with other research in the field. 

• I have made clearer the distinction between the three frames we used and the four components Entman suggested made up a frame. I have referenced a methodological paper we have written on this version of framing analysis, in lieu of including supplementary material. 

• I have explicated that Author 1 undertook the framing analysis and detailed the input Authors 2 and 3 had in the process. 

Page 13, lines 278-280: Stated that the researcher considered and cross checked the initial findings – was there any refinement here as a result of checking? Any discussions with the wider research team?

 I have detailed the roles Authors 2 and 3 took in cross-checking the initial findings, and consequences for refinement of the findings, and setting the scene for later interpretation. 

Results

Page 13, first paragraph of results: Would have been useful to provide a high level overview here, including the number of papers across each of the three sample years, and perhaps the key trends that you saw that you will unpack further. This would be more informative than simply explaining how the results are structured 

 We have adapted the introduction to the findings section, to more clearly map out the key ideas present in the section and prepare the reader for the content of the section. We also view the structure of the results as complex, so have included navigational information on this following the introduction to key findings. 

One thing that for me was missing from the findings was a consideration of the frequency with which diabetes was actually discussed in terms of an ‘epidemic’ across the years of the sample – how often was this term actually used? This is a difficult query to resolve, as the term ‘epidemic’ is not the only way that media content can conjure up the notion of an epidemic. This is a benefit of qualitative analysis, in that it can help identify nuanced or latent meanings such as ‘timebomb’ or ‘avalanche’ or ‘impending catastrophe’ that convey the idea of an epidemic. The framing analysis in particular enabled a systematic view of these terms, and I have directly stipulated more frequently the emergence and naturalisation of these terms. 

Page 13, lines 285-286: Need more detail to explain the content of the tables – state that shading refers to salience – but not clear if darker means ‘more salient’ or how salience was judged – this needs to be explained somewhere. Would perhaps be helpful to expand these tables to include an example in each cell.

 We have added in more information around shading to the introduction of the results section. We view that including an example in each cell would detract from the value of using the rubric cells to convey multiple ideas where they exist, in a snapshot – as such we have decided not to make this revision. 

Page 13, lines 300-301: Stated that small sample in 1993 suggests that diabetes was not as newsworthy – but related to my point above, that without a search term of ‘diabetes’ on its own (not in conjunction with obesity or epidemic) you can’t really make this statement – there may well have been articles in 1993 that talked about diabetes but not in the ‘linked’ way that your search terms targeted and that therefore may have been missed by your search.

 Thank you for this important point. We have revised the paper as a whole, and this detail, so that it more clearly positions our new knowledge about UK news presentations of diabetes and its’ epidemic in relation to obesity and the obesity epidemic. 

Page 16, lines 353-356: This paragraph on medical framing may need some nuancing to make it clear why this falls into the ‘modifiable’ rather that ‘non-modifiable’ risk factors

In a number of the quotes that you use to support your descriptions of ‘lifestyle/behavioural framings’ of diabetes in the 2001 and 2013 samples, the data doesn’t actually explicitly mention lifestyle or behaviour and this seems to be implied by the author. For example, lines 363-368 the description focuses on role of lifestyle but the quote only mentions obesity as a cause of T2DM, without reference to lifestyle or behaviour. Similarly, the quote on page 20, lines 472-474 doesn’t mention lifestyle but is used in apparent support of the statement that there was a perception that diabetes and obesity can be addressed through lifestyle

 Thank you for this detail. We have tightened up our use of ‘lifestyle’ vis a vis other modifiable risk factors to reference each in relation to how they are portrayed in the data. We later summarise the overarching interpretation around this in relation to lifestyle. 

I have moved up the comment about medical framing to the introductory section for 2001 so this is not confusing for the reader. 

Page 16, lines 370-374: This seems to pretty much just repeat what the description and quote in the previous lines was saying

 I have revised this content to make clearer the expansion from obesity to overweight as the key point in this part. 

Throughout the results it is not always clear which framings the data you are talking about fall into. It would perhaps be useful to have a few sentences at the start of the section for each year just summarising the key patterns in the data that are represented in the figure, before delving down into more detail and examples.

 I have synthesised some of the findings from 2001 and 2013 into respective orienting paragraphs – in order for each subsection to remain as concise as possible and minimise repetition. 

For 1993, because the introduction to the results section outlines significant differences between 1993 and 2001/2013 samples (i.e. in response to Reviewer #2), it seems logical that this will remain fresh for readers when they arrive to the 1993 subsection, as such it doesn’t need its own introduction. 

I would also perhaps have liked to see a reflection on whether there were differences across the three kinds of newspapers in how they portrayed diabetes?

 We have included a section on this at the end of the discussion section, which also comments on the differences noted between types of articles. 

Page 19, line 433-434: Stated that “diabetes increasingly presented as preventable through individual action and choice alone” – but the quotes that you present suggest a more nuanced understanding – i.e. that these things reduce risk, not that there are the only thing to do.

 This is a helpful point – thank you – on revisiting we have changed this to reflect this more nuanced understanding (in this instance and throughout the paper). 

Discussion

Findings are situated well within the broader context and literature although, I would expect to see some consideration of the limitations of the study here. 

 Thank you for this feedback. I have collected the information on limitations from within the paper and focused it into a section between the discussion and the conclusion.

---

## [Editor Report · Decision Letter 1]

13 Nov 2019

Monitoring the ‘Diabetes Epidemic’: A Framing Analysis of UK Print News 1993-2013

PONE-D-19-22797R1

Dear Dr. Foley,

We are pleased to inform you that your manuscript has been judged scientifically suitable for publication and will be formally accepted for publication once it complies with all outstanding technical requirements.

With kind regards,

Quinn Grundy, PhD, RN

Academic Editor

PLOS ONE

Additional Editor Comments (optional):

Thank you again for giving us the opportunity to review this manuscript - I am so pleased to increase the representation of critical, qualitative research in PLoS ONE.

PLoS One does not provide copy editing services. When you submit the revision complying with technical requirements, I suggest you make the following copy edits:

Spell out UK in title as United Kingdom

Spell out first instances of UK and US in text.

Line 135: Comma after "US", New York Times italicised

Line 162: Comma after "in a spiraling fashion"

Italicise the names of the newspapers sampled (eg line 181).

In Table 1, consider making the final row "Sources" a footnote, or adding the citation to the column header plus a footnote.

The quotation on lines 437-438 seems to be missing some punctuation.

Line 696, Comma after "In 1993"

Line 815, Commas after "Through linking diabetes with obesity"

Please include Legends for Figure 3, 4, and 5 that spells out the acronyms (BGLs, GT)
---

## [Editor Report · Acceptance letter]

9 Jan 2020

PONE-D-19-22797R1 

Monitoring the ‘Diabetes Epidemic’: A Framing Analysis of UK Print News 1993-2013 

Dear Dr. Foley:

I am pleased to inform you that your manuscript has been deemed suitable for publication in PLOS ONE. Congratulations! Your manuscript is now with our production department. 

With kind regards,

on behalf of

Dr. Quinn Grundy 

Academic Editor

PLOS ONE